# A New One-Tube Reaction Assay for the Universal Determination of Sweet Cherry (*Prunus avium* L.) Self-(In)Compatible MGST- and S-Alleles Using Capillary Fragment Analysis

**DOI:** 10.3390/ijms24086931

**Published:** 2023-04-08

**Authors:** Jana Čmejlová, František Paprštein, Pavol Suran, Lubor Zelený, Radek Čmejla

**Affiliations:** Research and Breeding Institute of Pomology Holovousy Ltd., Holovousy 129, 508 01 Hořice, Czech Republic

**Keywords:** S-allele, M locus-encoded glutathione-S-transferase, MGST, sweet cherry, self-compatibility, self-incompatibility, fragment analysis, molecular marker assisted breeding, MAS

## Abstract

The sweet cherry plant (*Prunus avium* L.) is primarily self-incompatible, with so-called S-alleles responsible for the inability of flowers to be pollinated not only by their own pollen grains but also by pollen from other cherries having the same S-alleles. This characteristic has wide-ranging impacts on commercial growing, harvesting, and breeding. However, mutations in S-alleles as well as changes in the expression of M locus-encoded glutathione-S-transferase (MGST) can lead to complete or partial self-compatibility, simplifying orchard management and reducing possible crop losses. Knowledge of S-alleles is important for growers and breeders, but current determination methods are challenging, requiring several PCR runs. Here we present a system for the identification of multiple S-alleles and MGST promoter variants in one-tube PCR, with subsequent fragment analysis on a capillary genetic analyzer. The assay was shown to unequivocally determine three MGST alleles, 14 self-incompatible S-alleles, and all three known self-compatible S-alleles (S3′, S4′, S5′) in 55 combinations tested, and thus it is especially suitable for routine S-allele diagnostics and molecular marker-assisted breeding for self-compatible sweet cherries. In addition, we identified a previously unknown S-allele in the ’Techlovicka´ genotype (S54) and a new variant of the MGST promoter with an 8-bp deletion in the ´Kronio´ cultivar.

## 1. Introduction

Sweet cherry (*Prunus avium* L.), one of the most important fruit crops in the temperate climatic zone, is primarily self-incompatible (SI), though self-compatible (SC) cultivars also exist. Self-incompatibility is a system developed by many plants to avoid self-pollination in order to increase gene flow in their populations but may have negative impacts on crop production and breeding [1,2]. Sweet cherry self-incompatibility also has far-reaching consequences for their commercial cultivation. It was confirmed already in the late 1940s that not all sweet cherry cultivars can pollinate each other [3]. Therefore, a suitable pollinizer had to be experimentally identified for each cherry cultivar to attain maximum fruit production. However, growing two different cultivars in one orchard can lead to complications during harvest as both cultivars may ripen at different times or bear fruits with significant differences in quality, shape, or color, and cannot be mixed for consumers. Therefore, modern sweet cherry breeding trends have focused on the production of SC cultivars to considerably simplify orchard management and reduce the risk of losses/reductions of crop yields.

The necessity of cross-pollination due to self-incompatibility in sweet cherry is based on the existence of a multiallelic S-locus carrying two genes in a non-recombining haploblock: S-allele-specific ribonuclease (S-RNase) participating in the female (pistil) part of self-incompatibility, and S-haplotype-specific F-box (SFB) ensuring the male (pollen) part of self-incompatibility [4,5]. S-alleles are particular variants of an S-locus that differ in their DNA sequence and have been numbered consecutively, S53 being the latest [6]. Successful pollination and fertilization occur when an S-allele in a haploid pollen grain genome is different from both S-alleles of the diploid pistil (exceptions are SC S-alleles). However, the exact mechanism preventing self-pollination is still unknown. According to a currently acknowledged model [7,8,9], a “general inhibitor” inactivates all S-RNases permeating into the embedding pollen tube from pistil tissues. This general inhibitor has been named the Skp1-cullin1-F-box protein complex (SCF) and is composed of several SLF-like proteins (PavSLFLs), SFB-like protein 2 (PavSFBL2), M-locus-encoded glutathione S-transferase (MGST), DnaJ-like protein, and other minor proteins [7,8,9]. In a compatible pollen tube, S-RNase is probably recognized by SLFL/SFBL proteins that mediate its polyubiquitination leading to its subsequent degradation by proteasomes. It is presumed that the SFB protein of the pollen tube is able to selectively recognize S-RNase from the same S-locus and protect it against polyubiquitination by the SCF general inhibitor. Active S-RNase can thus exert its function—to destroy RNAs in an embedding pollen tube. This induces the end of pollen tube growth already in the stigma and prevents self-pollination. The role of other proteins from the SCF protein complex is still unknown [7,8,9]. 

Regardless of the exact mechanism of pollen grain rejection by a pistil, decisive roles for compatibility in cherries have so far only been described for S-RNase, SFB, and MGST. An inactivating mutation in S-RNase leading to self-compatibility was found in the closely related sour cherry (*Prunus cerasus* L.) which is a natural hybrid between *Prunus avium* L. and *Prunus fruticosa* Pall. (e.g., in [10]). Currently, all fully SC sweet cherry genotypes carry an inactivated SFB protein, and the mutated version of the particular S-allele is marked by an apostrophe. S-alleles S3′ and S4′ were generated by irradiation [11], with an extensive deletion in SFB of an unknown range induced in S3′, but only a 4-nucleotide deletion in S4′ leading to a frameshift and premature stop codon [12]. On the other hand, the S5′ allele found in several Sicily cultivars has a natural origin and contains a point mutation in the SFB gene leading to premature stop codon [13]. The importance of the MGST gene, part of the SCF general inhibitor, for self-(in)compatibility in sweet cherries has also been described. It was shown that the at-least partial self-compatibility observed in some Spanish cultivars was due to a decrease in MGST expression caused by an insertion of a transposon-like sequence in the promoter region of this gene [14,15].

The first laboratory attempts to elucidate sweet cherry incompatibility were biochemical methods [16,17]. Soon afterwards, molecular methods were employed, enabling the analysis of S-alleles from any tree tissue containing DNA. Allele-specific primers were designed to identify the first six S-alleles described [18]. However, more and more S-alleles were identified, and the need for universal primers for S-allele detection quickly became a priority. Due to differences in the length of introns, universal degenerated primers could be designed in exons of the S-RNase and SFB genes, to be used for PCR and subsequent fragment analysis. A pair of PaConsI-F and PaConsI-R2 primers [19,20] amplifies a region of the first S-RNase intron resulting in PCR products up to 500 base pairs (bp) in length that could be distinguished easily in a fragment analysis run using a capillary genetic analyzer. The number of S-alleles identified continued to grow, however, and PCR products from these primers for some alleles have the same or nearly the same size, for example, S2, S7, and S12 [20,21,22]. Although a capillary genetic analyzer is able to discriminate fragments differing by 1 nucleotide (nt), doubts may arise concerning fragment identity, and it is therefore recommended to run a control sample with known S-alleles in parallel. Furthermore, the S13 allele is nearly unamplifiable by these primers [12,23]. The primers PaConsII-F and PaConsII-R [19] were therefore developed to detect length polymorphisms of the second S-RNase intron, but PCR products must be analyzed by agarose gel electrophoresis due to their sizes (up to nearly 2.5 kbp), which reduces the resolution of the analysis. Moreover, some S-alleles were hard to amplify with these primers as well (for example S5, S13, and S16) [19]. Other universal primers were designed for amplification of an SFB intron: however, (i) they are degenerated to such a degree that a carefully optimized touch-down PCR temperature profile must be applied, and (ii) PCR products for several S-alleles have the same length, e.g., S1, S4, S9, and S19; S5 and S17; S10 and S22; and S13 and S14, respectively [21]. To overcome these problems, S-allele-specific primers were developed [18,19,24], and used in a separate PCR run either alone or in a combination with universal primers (for example [25,26]).

However, correct S-allele determination may be further complicated by the presence of SC S-alleles that cannot be (easily) distinguished by these universal primers from their SI counterparts, and a separate analysis has to be done to detect underlying mutations in order to discriminate them. A PCR system for S3′ identification was described by Sonneveld et al. [12], while nested PCR with a subsequent detection either by denaturing polyacrylamide gel electrophoresis or by agarose gel electrophoresis after cleavage by a restriction endonuclease was presented for the S4′ allele by Ikeda et al. [27]. Later, Muñoz-Espinoza et al. (2017) developed a high-resolution melting analysis for S4′ allele identification [28]. The S5′ allele may be differentiated from its S5 counterpart by fragment analysis of an SSR marker located in the second S-RNase intron [13]. The presence of a transposon-like sequence in the MGST promoter may be detected by PCR described by Ono, 2018 [15]. In summary, there is currently no versatile, simple, one-tube reaction protocol that would allow the identification of all individual MGST and S-alleles, especially in case the S-allele composition of parents is unknown, e.g., in gene pools or open pollination breeding.

Knowledge of the S-allele composition in sweet cherry cultivars is essential for both growers and breeders because many commercial cultivars are unable to pollinate each other. To date, 22 different SI S-alleles have been described in commercial sweet cherry cultivars—S1, S2, S3, S4, S5, S6, S7, S9, S10, S12, S13, S14, S16, S17, S18, S19, S21, S22, S24, S27, S30, S37 [29]. In the list of 1,483 different cultivated sweet cherry cultivars, S-alleles S3, S1, S6, S4, S9, S2, S5, and S13 were the most common, representing more than 87% of S-alleles (ordered by frequency; S-alleles with more than 100 different genotypes are specified). A total of 63 so-called incompatibility groups of sweet cherry cultivars carrying the same S-allele combination have been described, and some of them comprise more than 100 cultivars that cannot successfully pollinate each other at all [29]. On the other hand, 26 sweet cherries have a unique combination of S-alleles, and can thus be considered universal pollinizers for all other cultivars [29].

From an agronomical point of view, more competing SC cultivars have been developed relatively recently; the first commercially successful cultivar was ‘Stella’ coming from a ‘Lambert’ (SI) and ‘JI2420’ (SC) crossing [30]. Currently, 91 SC genotypes are recorded in the cultivated sweet cherries list [29], containing mainly the S4′ allele. The market potential of known SC cultivars is, however, still dubious, and their participation in worldwide sweet cherry production remains low as they usually do not meet the expected parameters of SI cultivars. New SC genotypes are therefore produced every year by breeders in an effort to get “improved” SC genotypes, for example with larger and firmer fruits ripening earlier or later in the cherry season. Identification of SC genotypes from crossings with SC cultivars by classical field tests of self-compatibility is laborious, and can only be accomplished after several years of cultivation when a tree first bears fruits. In contrast, molecular methods enable the identification of the S-allele genotype in several-week-old seedlings, thus saving time, labor, and costs for the maintenance of non-perspective seedlings.

The aim of this work was to develop a universal, simple, and rapid one-tube assay that would allow the identification of known sweet cherry S-alleles, both SI and SC, together with MGST promoter alleles, using fragment analysis run on a capillary genetic analyzer to simplify, speed up and cheapen the analysis of sweet cherry compatibility. The assay is especially suitable for both routine compatibility determinations and molecular marker-assisted selection (MAS) focused on SC sweet cherry breeding.

## 2. Results

### 2.1. Assay Design

To provide a comprehensible assay, the developed system was divided into several subsystems that were optimized to work together in a one-tube format—a subsystem for the universal detection of SI S-alleles; a subsystem for the identification of SC S-alleles; a subsystem for the determination of MGST alleles; and a control fragment of the *PaveIF-1A* gene that was used for two purposes—specifically for the determination of genotypes containing special combinations of SC alleles, and generally as internal quality control for the whole assay. For the easy interpretation of results, fluorescent channels were dedicated as follows: blue (6-FAM dye) for the universal detection of S-alleles including SC S-alleles determination; green (VIC dye) for the detection of MGST alleles; black (NED dye) for the detection of the *PaveIF-1A* gene control fragment; and red (PET dye) for the specific detection of selected S-alleles. An overview is presented in Figure 1.

The developed system for the S-allele determination in one reaction is based on an optimized universal PaConsI primer pair used for the amplification of the S-RNase intron 1 since it is expected to also amplify unknown/untested S-alleles. These primers were supplemented with primers specific for individual S-alleles where necessary to discriminate S-alleles with the same or very similar PaConsI fragment length. Specific primers were designed to distinguish SC and SI S-alleles, and finally, the MGST allele combination is determined using allele-specific forward primers. Special attention was paid to the correct interpretation of particular allele combinations (e.g., S3S3′, combinations of two of the same SC S-alleles) by the use of a quantitative analysis based on PaveIF-1A-specific fragment signal height. Primer characteristics are shown in Table 1.

### 2.2. Subsystem for the Detection of SI S-Alleles

As starting material, nearly 340 accessions from a germplasm collection (Appendix A) were analyzed with the published universal PaConsI primer pair [19,20] by fragment analysis, and the identity of individual peaks was verified by sequencing. Most of the cultivars displayed two S-alleles, but sometimes only one S-allele was observed (some of these cultivars were previously identified as containing the S13 allele). S–RNases amplified with the published universal PaConsI primer pair showed different signal intensities after fragment analysis, implying primers had to be optimized before use in a multiplexed PCR for the detection of all S-alleles. Since not all S-alleles have known sequences of binding sites for the universal PaConsI primer pair, a new universal primer pair (S-RNase-seq-F and S-RNase-seq-R) was designed flanking a supposed PaConsI amplicon to check for potential primer mismatches. While the reverse PaConsI-R2 primer binding site was found conserved, analysis of the forward PaConsI-F binding sites revealed a number of mismatches that could potentially lead to an inefficient amplification of some alleles (Figure 2). Based on the results, five new PaConsI-F-class primers shortened by one nucleotide at the 5′ end compared to the published PaConsI-F primer were designed to avoid a degenerated nucleotide at this position (PaConsI-CTTC-F; PaConsI-GTTC-F; PaConsI-CTCC-F; PaConsI-CTTT-F; and PaConsI-CGTC-F; in general PaConsI-xxxx-F) and were supplemented with the original PaConsI-R2 fluorescently labeled by 6-FAM to cover all S-alleles.

This combination of primers was successfully used for the amplification of all previously identified alleles. Amplification of S13 with the PaConsI-F primer tailored for this allele repeatedly failed in all cultivars known to contain S13 allele, thus a specific primer for S13 was finally designed in intron 2 (S-RNase-S13-In2-F) to be used with the common primer designed in exon 3.

Another cultivar, ‘Techlovicka’ (synonyms: ‘Techlovicka I’ or ‘Ziklova’), gave only an S4 allele using the PaConsI primer pair. In addition, specific determination of S13 failed, implying that a new allele not amplifiable with either universal PaConsI primers or S13 specific primers had to be present in this genotype. The new S-allele was amplified and sequenced using the new universal S-RNase-seq-F and S-RNase-seq-R primer pair, and the full genomic sequence was then compiled from Sequence Read Archive (PRJNA813711, library name SA777). The identified S-allele was numbered S54 (after personal communication with Drs. Schuster & Schröpfer) and showed 98.3% nucleotide sequence identity (97.3% protein identify) with the *Prunus virginiana* S2 allele (GenBank No: JQ627790; 518 nt sequence; 147 amino acids) (Appendix A). The phylogenetic comparison revealed that S54 (deposited in the GenBank database under No: OQ555802) is related to *P. avium* S27; *P. speciosa* S34; and *P. cerasus* S35 (Appendix A). As the PaConsI universal primers did not work for this allele, a specific primer set was designed for separate detection in the red channel (PET-labeled S-RNase-S54-F + S-RNase-S54-R).

Besides specific primers for S54 and S13, primers specific for S1, S2, S6+S24, S7, S9+S22, and S12 had to be designed since the observed fragment sizes obtained with PaConsI-xxxx-F and PaConsI-R2 were the same or differed by only 1 nt (see Table 2 below for expected amplified fragment lengths). Forward primers rendering specificity (S-RNase-S#-In2-F) were located in S-RNase intron 2 and were complemented by a common reverse PET-labeled primer S-RNase-Ex3-R lying in the conserved region of S-RNase exon 3 (Appendix A).

### 2.3. Subsystem for the Identification of SC S-Alleles

As all SC alleles have a causal mutation in the SFB gene, distinguishing them from their SI counterparts by S-RNase fragment analysis may be difficult (S5 vs S5′—1 nt length difference) or even impossible (same length: S3 vs S3′; S4 vs S4′). An alignment of all known cherry sweet SFB genes was done to design S-allele-specific primers. In the case of the S3′ allele, self-compatibility is caused by an uncharacterized deletion in the SFB gene. To distinguish the S3 allele from the S3′ allele, a PCR primer pair (newly designed 6-FAM-labeled PaSFB3-short-R together with PaSFB3-F published by Sonneveld et al. [12]) was used to specifically amplify a fragment within the S3 SFB gene that is deleted in the S3′ allele, i.e., the presence of both S3 specific fragments (227 nt: S-RNase origin, 122 nt: SFB origin) indicates the wild-type S3 allele, while the presence of only one 227 nt fragment originating from S-RNase implies the S3′ SC S-allele.

The S4′ allele is characterized by a 4bp deletion that could be easily recognized from the wild-type allele by fragment analysis using PaSFB4+4′-F and PaSFB4+4′-R primers encompassing the deletion (forward primer labeled by 6-FAM). 

Special attention was given to distinguishing the S5 and S5′ alleles, as a point mutation in the S5′ SFB gene is responsible for self-compatibility, and as such the two alleles cannot be distinguished by fragment length analysis of this region. The primer PaSFB5-F was therefore developed to amplify only the S5 SFB wild-type allele—the primer 3′ end is located at the site of the mutation, plus a destabilizing mutation near the primer 3′ end had to be introduced to ensure 100% specificity. Thus, the S5 and S5′ alleles can be primarily determined by the PaConsI-class primers (S5 fragment 385 nt; S5′ fragment 384 bp), and the identity of the S5 allele is further corroborated by the presence of an 83 nt fragment originating from the SFB gene (primers PaSFB5-F+6-FAM-labeled PaSFB5+5′-R).

### 2.4. Subsystem for the Determination of MGST Alleles

The ‘Cristobalina’ cultivar and some other Spanish cultivars are known for partial self-compatibility due to an insertion of 1848 nt in the promoter region of the *MGST* gene. To distinguish the wild-type allele (MGSTwt) from the SC MGST allele (MGSTins), the MGST-TE-out-F primer was located upstream of the insertion, and the MGST-TE-in-F primer was placed in the insertion. A common VIC-labeled MGST-TE-out-R primer [15] for the amplification of both alleles was used to reliably discriminate each allele in the green channel, which was used for the MGST analysis only.

During the course of testing, a third allele was revealed in the ´Kronio´ cultivar that possessed an 8 nt deletion on the wild-type background in the amplified region (MGSTdel). Sequence analysis of MGST alleles is presented in Appendix A. This allele could be also clearly assigned in fragment analysis (Appendix A).

### 2.5. Final Multiplexing and Data Interpretation

Every single fragment individually amplified with newly designed primers was verified by sequencing as well as by fragment analysis. The specificity of primers was tested using cultivars with different allele compositions to exclude false positivity. In all cases, primers proved to be 100% specific. In the next step, verified subsystems were multiplexed to obtain a one-tube PCR format, containing all 27 primers. PCR conditions as well as concentrations of each primer were adjusted (Table 1) to obtain approximately the same signal for all fragments (Figure 3).

Finally, matrices for the clear interpretation of findings were created—each allele is represented by a unique (combination of) fragment(s) (Table 2), and each fragment is associated with a particular allele(s) (Table 3).

The assay was able to successfully determine 14 SI S-alleles (S1, S2, S3, S4, S5, S6, S7, S9, S12, S13, S13, S22, S24, and newly identified S54), three SC S-alleles (S3′, S4′, and S5′), and three MGST alleles (Appendix A). It is expected, however, that all previously known S-alleles identified with PaConsI primers should be detected and discriminated against as well. However, those genotypes were not available for analysis. 

DNA isolation kits were in general expected to work comparably, thus only two kits were tested (Exgene Plant SV isolation kit, GeneAll, as a reference kit; and NucleoSpin Plant II, Macherey-Nagel, as the tested kit) using a limited number of samples (n = 10) with identical results obtained. As for PCR reagents, two additional master mixes (PCR Master Mix (2X), ThermoFisher Scientific, Waltham, MA, USA, and qPCR 2x Blue Master Mix, Top-Bio, Vestec, Czech Republic) intended for general PCR work were tested and compared with a Phusion Flash High-Fidelity PCR Master Mix (ThermoFisher Scientific, Waltham, MA, USA) as a refere nce. Concentrations of primers and DNA were the same in all tests, with only reaction volumes and temperature profiles modified following manufacturers’ recommendations. The number of cycles was adjusted to observe all expected peaks. Both the PCR Master Mix (2X) and qPCR 2x Blue Master Mix successfully amplified all expected fragments; however, some of them were double-peaked, but this did not hinder correct allele calling (Appendix A).

In general, the results of fragment analysis were readily comprehensible and clearly indicated the alleles present in a sample. However, special attention was paid to the correct interpretation of fragment analysis outputs involving S3′, S4′ or S5′ allele only, since the same self-compatible alleles can be combined; and to the output when the S3 allele is solely detected as in S3S3′ genotype. In theory, these four SC allele combinations may be masked by another genotype containing a yet unknown S-allele that could not be detected by the PaConsI-class primers (denoted as S?): S3S3′ can be misinterpreted as an S3S? genotype; S3′S3′ as S3′S?; S4′S4′ as S4′S?; and S5′S5′ as S5′S?. If only an S3, S3′, S4′, or S5′ allele is detected, further analysis is therefore required to decide about the second S-allele. For this purpose, the fragment of the *PaveIF-1A* gene was added to help with data renditions. In all samples tested, PaveIF-1A was represented by one allele, implying it can be used as a general calibrator for a diploid sweet cherry genome to which the signal (height) of other fragments can be normalized.

As no appropriate genotypes with S? were available, it was necessary to mimic these theoretical situations. S3S3′ and S3′S3′ were compared with S3Sx and S3′Sx genotypes, respectively, where Sx means any detected S-allele except for S3′; and S4′S4′ was put in contrast to S4′Sy genotypes, where Sy stands for any detected S-allele except for S4′. The S3S3′ combination would be expected to exhibit a diploid signal for an S3 S-RNase fragment (227 nt; blue channel) and a haploid signal for a wild-type S3 SFB fragment (122 nt; blue channel), while S3Sx would be expected to produce a haploid signal for both fragments (Figure 4). Thus, after normalization to the control diploid PaveIF-1A fragment (118 nt; yellow channel), the signal intensity (height) of a fragment directly indicates its copy number (Table 4). The height ratio of the common S3+S3′ S-RNase fragment (227 nt) to the wild-type-specific S3 SFB fragment (122 nt) can also be used as an alternative.

The same analytical pipeline may be used for SC genotypes containing the SC allele in two copies: the S3+S3′ S-RNase fragment (227 nt; blue channel) was normalized to the PaveIF-1A fragment (118 nt; yellow channel) for S3′S3′, resulting in a highly statistically significant difference between S3′S3′ and S3′Sx (Figure 5, Table 5). Alternatively, a diploid MGSTwt signal can be used as a control gene, since the ratio of both control genes was nearly the same in both genotype groups. Normalization of the S3+S3′ S-RNase fragment (227 nt; blue channel) to MGSTwt (140 nt, green channel) showed also a highly significant difference as expected (Table 5). 

Similarly, for S4′S4′genotypes, the S4+S4′ S-RNase fragment (443 nt; blue channel) and S4′ SFB fragment (177 nt; blue channel) showed statistically significant differences between both genotype groups after their normalization to the PaveIF-1A fragment (118 nt; yellow channel), and there was no significant difference in the ratio of these peaks between groups (Figure 6, Table 6). 

In the case of the S5S5′ S-allele combination, the presence of both alleles can be readily distinguished since they differ by 1 nt in size using PaConsI-F-class primers (as no S5S5′ genotype was available for analysis, the situation was demonstrated by a 1:1 mixed sample of ‘Kronio’ (S5′S6) and ‘Uriase de Bystrita’ (S5S12) in Appendix A). The S5′S5′ genotype was also not available for study; however, deciphering its presence would follow the rules presented above for the S3′S3′ and S4′S4′ genotypes. Using this approach, dubious SC genotypes can be clearly distinguished from other genotypes.

### 2.6. Assay Verification

The whole germplasm collection at the Research and Breeding Institute of Pomology Holovousy containing 339 genotypes (Appendix A) as well as ‘Cristobalina’ (for MGST self-compatible testing) and ‘Kronio’ (for S5′ testing) cultivars were re-analyzed using the final kit. In all genotypes, presumed MGST alleles were confirmed, and two S-alleles were detected in the expected combinations. The frequency of individual S-alleles in the germplasm collection is depicted in Figure 7.

The most frequent SI S-alleles were S3 (181×) and S1 (117×); in contrast, S7, S16, S22, and S25 occurred only rarely (1–4×). A new S54 allele was identified that was only present in the source cultivar (´Techlovicka´). Special attention was paid to SC alleles: S3′ was identified in five genotypes, S4′ was the most frequent SC allele (23×), and S5′ was found only in the donated ´Kronio´ cultivar. As for the MGST promoter variants, only MGSTwt alleles were observed, the other two alleles were found only in the donated cultivars (the MGSTins allele in ´Cristobalina´, and the MGSTdel allele in ´Kronio´) in combination with the wild-type allele.

The kit was also used to test material originating from a breeding program focused on self-pollinating sweet cherry genotypes involving a different combination of S3′ or S4′ parents. Blind samples (n = 295) were provided to the laboratory, and in all cases the allele combinations were unambiguously detected, confirming the utility of the assay for routine testing. Self-fertility was proved by field testing in some cases.

## 3. Discussion

Knowledge of S-alleles is important in all cultivation of sweet cherry cultivars, but the determination of S-allele composition has so far been laborious and time-consuming, especially when no information about presumptive S-alleles is available (i.e., in cases of open pollination, undesirable pollination with an unintended pollen grain, or in gene pools). Currently used systems usually rely on several PCRs, mainly exploiting intron length polymorphisms between individual S-alleles [19,20,21,25,26]. Another approach enabling the unambiguous assignment of S-alleles is PCR using allele-specific primers, for example, the 13 primer pairs specific for S1 to S16 [18,19], or allele-specific primers for S17 to S19, S21/25, S34, and S37 [24]. SC alleles have to be deduced from the results of a separate analysis of an underlying mutation [12,13,27,28], complicating further the analysis of S-alleles present, since this is usually run in several steps. 

The aim of the present work was therefore to develop a simple assay that would enable the recognition of sweet cherry self-(in)compatibility and potential pollinating cultivars in a single one-tube PCR step followed by fragment analysis on a capillary genetic analyzer. This newly developed system was able to successfully recognize all common S-alleles usually occurring in commercially cultivated cultivars. Moreover, it is also expected to recognize all other identified S-alleles, since it employs universal primers flanking the first S-RNase intron that were originally used to identify them. Some rare S-alleles were not yet analyzed by the assay; however, the expected length of respective PCR fragments may be deduced from published sequences, and their relative size in fragment analysis can be estimated based on the labeling: 6-FAM- and PET-labeled fragments were about 3–7 nt and 0–2 nt, respectively, shorter than expected (Table 2).

To achieve the best fragment analysis outputs from such a multiplex reaction, the widely used PaConsI primer pair [19,20] had to be substantially improved as the published primer degeneration did not cover all S-alleles with 100% homology when compared to newly sequenced primer binding sites (Figure 2). As a consequence, S-alleles with less than 100% homology to the degenerated primer were under-amplified in such a complex system, consistently exhibiting peaks with lower intensity in the fragment analysis. The previously published degenerated forward primer PaConsI-F [19] was therefore replaced by five single primers, which made it possible to optimize the concentration of individual primers. Thus, the PaConsI-CTTT-F primer with the lowest melting temperature was used in a much higher concentration than the other PaConsI-xxxx-F primers. The previously published PaConsI-F primer was also shortened at its 5′end by one nucleotide to avoid further degeneration. The reverse primer was used unchanged and was fluorescently labeled to detect all fragments irrespective of the forward primer involved in the PCR amplification. Two exceptions were observed that were not amplified by the PaConsI primers. Although S13 S-RNase should be amplified from this primer pair (a forward primer with 100% homology was used), the presence of a short tandem repeat (TA)_22_ in the amplified region probably inhibited the PCR, as has been observed also by others [20,23]. It is therefore not surprising that no S13 amplification with this primer pair was observed in such a multiplex system. S54 S-RNase escapes amplification from PaConsI primers due to multiple mutations under the forward primer, but also a mutation under the reverse primer. 

The system was further supplemented by S-allele specific primers placed in the second S-RNase intron to verify S-alleles of similar/same fragment lengths amplified with the PaConsI-class primers and S13. To minimize the cost of the assay, these specific PCR primers were paired with a universal PET-labeled primer lying in S-RNase exon 3. The simultaneous use of S-allele specific primers in the PCR mix thus minimizes the necessity for additional control reactions for the determination of these S-alleles. S-allele specific primers for S54 were located in the first S-RNase exon and intron.

Primers able to specifically distinguish S3 and S3′, S4 and S4′, and S5 and S5′ alleles were included as well. Since SC genotypes can pollinate themselves, S-allele combinations containing the same two alleles can occur, making correct interpretations difficult. In this respect, a system was developed for the precise discrimination of S3S3′, S3′S3′, S4′S4′, and S5′S5′ genotypes (with a false determination chance < 10^−19^; Table 4, Table 5 and Table 6), using a normalized quantitation of an S-allele-specific fragment to a control PaveIF-1A gene fragment [31]. To our knowledge, this is the first assay capable of determining the S-allele combination in SC genotypes in one step, making it an excellent tool for sweet cherry breeding programs focused on the production of SC genotypes.

Finally, primers for the determination of partial self-compatibility caused by an insertion in the MGST promotor were also included in a separate green channel. A common VIC-labeled reverse PCR primer was used in combination with two allele-specific forward primers. In case of doubt about MGSTwt dosage in a genotype, normalization of the peak signal to PaveIF-1A can be done. 

Though the assay was designed to be universal, the number of primers can be changed according to the exact needs of the breeders; for example, the subsystem for the determination of MGST alleles may be omitted if a breeding program does not target it. Usage of the PaConsI-F primer instead of five PaConsI-xxxx-F primers is possible but not recommended, as some S-alleles with a lower homology to the PaConsI-F primer displayed a low signal.

To get reproducible results, the assay employs a couple of checks: (i) a standard amount of a DNA template input (20 ng, the A260/A280 ratio >1.8); (ii) a standard PCR procedure (e.g., using verified PCR reagents); (iii) an internal positive control (a fragment of the *PaveIF-1A* gene) that exhibits an expected signal height; (iv) alternatively, an MGST fragment can be used in case MGST is not the object of interest (i.e., homozygous MGSTwt); (v) a specific detection of S1, S2, S6+S24, S7, S9+S22, S12 alleles to corroborate findings based on the PaConsI universal primers; and most importantly, (vi) two S-alleles are always expected to be clearly detected, otherwise attention has to be paid to correctly interpret fragment analysis data. 

The whole system was intended to be low-cost, therefore, only eight primers were fluorescently labeled out of a total of 27 primers. Moreover, the PCR mix volume was downsized to 10 µL, reducing the cost of PCR reagents and consumables. The kit saves labor and shortens substantially the time of the analysis. It is recommended to admix all primers in a 10× primer master mix (typically for several hundred PCRs) and to test it using control samples containing all detectable S-alleles. The initial primer master mix verification is especially important for distinguishing the S-alleles based on the absence of a fragment (i.e., S3 vs S3′ and S5 vs S5′) to exclude PCR failure. This primer master mix pre-testing minimizes the number of controls needed in each fragment analysis run lowering further the analysis cost.

The assay was validated using a specific DNA isolation kit and PCR reagents, and it is up to the user to verify it under his/her own laboratory conditions. In general, DNA isolation kits providing sufficiently pure DNA (A260/A280 > 1.8) may be used. As for PCR reagents, there are many products on the market that differ greatly in their performance. We chose a Phusion Flash High-Fidelity PCR Master Mix as a reference because it had been shown before to successfully co-amplify up to 65 expected alleles in our study describing a new one-tube reaction kit for the SSR genotyping of apples [32]. However, general-purpose PCR reagents can also be used, though some optimization of primer concentrations/cycling profiles may be needed to obtain peaks of roughly the same intensity for easy interpretation (Appendix A). Generally, in search for an alternative polymerase, the following rules should be applied: (i) a polymerase should be robust to amplify both low and high complexity DNA templates; (ii) it should possess proofreading activity for reproducible results; (iii) it should produce blunt-ended fragments for clear interpretation; (iv) it should be hot-started to prevent the degradation of primers and template DNA during PCR setup.

During the development of the system, two new alleles were found. S54 was discovered in the ‘Techlovicka’ cultivar, a landrace of unknown origin from Eastern Bohemia. Interestingly, the S54 allele shares 98.3% nucleotide sequence identity with the *Prunus virginiana* S2 allele, indicating the ‘Techlovicka’ cultivar might originate from an inter-species cross. However, the ‘Techlovicka’ cultivar looks like a common sweet cherry without any sign of *Prunus virginiana* traits (phenotypic and phenology traits may be found in [33]), thus the introgression of the *Prunus virginiana* S2 allele sequence presents a puzzle warranting further investigation. The predicted S54 S-RNase protein is similar to other *Prunus* spp. self-incompatible S-alleles and self-compatibility was indeed corroborated by the finding that the self-pollination of this genotype repeatedly gave no fruits. The *SFB* gene was not analyzed. Since this S-allele is unique, the ‘Techlovicka’ cultivar could be used as a universal pollinizer; such universal pollinizers are rather rare [29]. Experiments focusing on the pollination capabilities of the ‘Techlovicka’ cultivar are ongoing.

The second new allele was identified in the MGST gene that was characterized as an 8 nt deletion in the promoter region of MGST in the ‘Kronio’ cultivar. However, the effect of this deletion on the self-incompatibility status is currently unclear, as the cultivar ‘Kronio’ is self-compatible owing to the presence of the S5′ allele. Hybrids containing only the MGSTdel allele but not containing the S5′ allele would be needed to verify the influence of this 8 nt deletion on MGST expression and hence the manifestation of self-compatibility.

The presented assay for the determination of self-(in)compatibility in only one reaction based on the identification of S-allele/MGST promoter allele combinations is suitable for the analysis of all sweet cherry genotypes, especially those with an unpredictable S-allele composition. It is also applicable for molecular marker-assisted selection (MAS), especially in breeding programs focused on the production of self-compatible sweet cherry cultivars or new universal pollinizers. MAS enables perspective seedlings to be selected after several weeks of growth, saving costs, labor, and the time necessary for the maintenance of undesirable SI genotypes in orchards until tests of self-compatibility can be realized (usually about 5 years). Savings from MAS applications can be large. At Washington State University it was estimated that MAS with various molecular makers (besides S-alleles, molecular markers for fruit size, firmness, and flavor were assessed) provided resource savings of almost USD80,000 per year when selecting 3000–3500 sweet cherry seedlings [34,35]. Tests for individual S-alleles can still be done to select a particular genotype; however, the newly developed system presented here offers a complex universal approach allowing the identification of all S-alleles in a one-tube PCR format, eliminating the need to determine the second S-allele in a separate reaction.

## 4. Materials and Methods

### 4.1. The Assay Design Workflow

The assay design was split into several steps that partially overlapped. First, fluorescent channels were assigned to targets, and second, a plan for the distribution of fragment sizes of target alleles was created for all channels to ensure that peaks would be unequivocally associated with the respective allele. 

The development continued in the following steps: (i) simplex identification and verification of common S-alleles by universal primers; (ii) pinpointing S-alleles that needed to be confirmed independently, and the design of specific primers; (iii) analyses of self-compatible S3′, S4′, and S5′ alleles, primer design and testing; (iv) special attention was paid to S-allele combinations that were complicated for interpretation (e.g., S3S3′, two of the same SC S-alleles in one genotype); (v) analysis of the MGST promotor, primer design; (vi) multiplexing and final tuning of primer concentrations; and (vii) verification of the assay using hundreds of genotypes.

### 4.2. Samples, DNA Isolation

A collection of 339 different genotypes of cultivated cultivars, landraces, and several wild-type sweet cherries originating from a germplasm at the Research and Breeding Institute of Pomology Holovousy was used for the system development and verification. Some of them were previously characterized by Patzak et al. [36]. Nearly 300 hybrid genotypes obtained within a breeding program potentially carrying S3′ or S4′ alleles were also analyzed. Shoots of ´Cristobalina´ (MGSTins allele) and ´Kronio´ (S5′ allele) were obtained for analysis as kind gifts from Dr. Mirko Schuster and Prof. Tiziano Caruso, respectively.

Shoot samples were collected in biological duplicates, and genomic DNA was isolated by an Exgene Plant SV isolation kit (GeneAll, Seoul, Korea) (or NucleoSpin Plant II, Macherey-Nagel, Dueren, Germany) according to the manufacturer’s instructions from 100 mg of phloem or leaves ground in liquid nitrogen. DNA with the ratio of absorbance at 260 nm and 280 nm > 1.8 was used for analysis.

### 4.3. The Design of Primers and Verification

Sequences of sweet cherry S-alleles (both S-RNases and SFBs) were obtained from the GenBank database (URL www.ncbi.nlm.nih.gov, accessed on 9 January 2020; Appendix A). Sequences were aligned using Geneious Prime^®^ software (version 2023.0.1; Biomatters Inc., Auckland, New Zealand). New universal primers S-RNase-seq-F (5′ CTAAGTATGGCGATGTTGAAATC) and S-RNase-seq-R (5′ GGGTTTGAATAATTACTTGGCC) were designed for the amplification of missing S-allele 5′ ends and used for sequencing. PCR was carried out under the following conditions: 20 ng of DNA template; 10 μL Phusion Flash High-Fidelity PCR Master Mix (ThermoFisher Scientific, Waltham, MA, USA), 0.25 mM each primer (S-RNase-seq-F and S-RNase-seq-R), PCR water up to 20 µL. PCR amplification was run on a C1000 PCR cycler (Bio-Rad, Hercules, USA) using the following profile—initial denaturation: 98 °C/30 s; cycling: 40× (98 °C/10 s, 58 °C/10 s, 72 °C/30 s); final extension: 72 °C/15 s. PCR products were analyzed on a 3% agarose gel and individual PCR fragments were cut out. Fragments were purified by an Expin Combo GP kit according to the manufacturer’s manual (GeneAll, Seoul, Korea). Purified PCR fragments were sequenced from both ends using a BigDye™ Terminator v3.1 Cycle Sequencing Kit (ThermoFisher Scientific, Waltham, MA, USA). Obtained sequences were verified by BLAST (Basic Local Alignment Search Tool; https://blast.ncbi.nlm.nih.gov/Blast.cgi). S-RNases with newly sequenced 5′ ends were aligned again, and primers for fragment analysis were designed using Geneious Prime^®^ software (version 2023.0.1; Biomatters Inc., Auckland, New Zealand).

Sequences of the MGST promotor were retrieved—the TE-like insertion from ‘Cristobalina’ was downloaded from GenBank, and the wild-type sequence of the promoter originating from the ‘Satonishiki’ cultivar was obtained from www.rosaceae.org (accessed on 31 August 2020; Appendix A). Sequences were aligned, the insertion was identified and primers were designed using the same software. 

Fragments amplified with all primer pairs used for the final fragment analysis were verified by sequencing 10 amplicons (if possible) from different genotypes. In the case of S5/S5′ alleles, primers SFB-S5/S5′-seq-F: GGTTTGCAGTTCTTGGGGT and SFB-S5/S5′-seq-R: GATTATACGATCACAATCACCCAA were used for verification of the S5′ inactivating point mutation. Primer specificity was also checked on cultivars with different known S-alleles used as negative controls. Amplification of the PaveIF-1A gene fragment served as a control, and previously published primers for *Prunus persica* L. [31] were adapted.

### 4.4. Fragment Analysis

In the first step, fragment analysis for individual S-allele/MGST/PaveIF-1A fragments was performed to identify the respective lengths. PCR was carried out under the following conditions: 2 μL DNA (10 ng/µL); 5 μL Phusion Flash High-Fidelity PCR Master Mix (ThermoFisher Scientific), 0,25 mM each primer, one of them fluorescently labeled, PCR water up to 10 µL. PCR amplification was run on a C1000 PCR cycler (Bio-Rad, Hercules, USA)—initial denaturation: 98 °C/30 s; cycling: 23× (98 °C/10 s, 60 °C/10 s, 72 °C/30 s); final extension: 72 °C/15 s. The PCR product (1 µL) was mixed with 15 µL Hi-Di Formamide and 0.5 µL GeneScanTM 600 LIZ^TM^ dye size standard (both ThermoFisher Scientific, Waltham, MA, USA). Fragment analysis was run on an AB3500 genetic analyzer (ThermoFisher Scientific, Waltham, MA, USA) after 2 min denaturation at 95 °C, and results were evaluated with GeneMapper software, version 5.0 (ThermoFisher Scientific, Waltham, MA, USA). 

After individual fragment analyses, all primers were multiplexed to be used in a one-reaction format, and analysis was repeated under the same conditions with the number of PCR cycles increased to 24. Concentrations of individual primers were subsequently adjusted to obtain approximately the same signal height for each peak. For routine work, all primers were premixed into a 10× primer master mix in such a way that the concentration of each primer was 10-times higher than indicated in Table 1. PCR conditions were as follows: final volume 10 µL; 10× primer master mix 1 µL; DNA 2 µL (10 ng/µL); water 2 µL; Phusion Flash High-Fidelity PCR Master Mix 5 μL; Cycling profile: initial denaturation 98 °C/30 s, cycling 24× (98 °C/10 s, 60 °C/10 s, 72 °C/30 s) final extension 72 °C/15 s.

Alternatively, PCR Master Mix (2X) (ThermoFisher Scientific, Waltham, MA, USA) and qPCR 2x Blue Master Mix (Top-Bio, Vestec, Czech Republic) intended for general PCR work were also tested following the manufacturers’ recommendations. For PCR Master Mix (2X): final volume 20 µL; 10× primer master mix 2 µL; DNA 2 µL (10 ng/µL); water 6 µL; PCR Master Mix (2X)10 µL; Cycling profile: initial denaturation 95 °C/3 min, cycling 29× (95 °C/30 s, 60 °C/30 s, 72 °C/30 s), final extension 72 °C/5 min. For qPCR 2x Blue Master Mix: final volume 20 µL; 10× primer master mix 2 µL; DNA 2 µL (10 ng/µL); water 6 µL; 2x PCR Master Mix 10 µL; Cycling profile: initial denaturation 94 °C/5 min, cycling 29× (94 °C/10 s, 60 °C/10 s, 72 °C/20 s), final extension 72 °C/7 min.

### 4.5. Statistical Analysis

Fragment analysis for the statistical evaluation of some specific S-allele combinations (S3S3′, S3′S3′, and S4′S4′) was run with 30 different biological/technical duplicates in each group. An F-test of the equality of variances and a t-test was done in MS Excel.

## Figures and Tables

**Figure 1 ijms-24-06931-f001:**
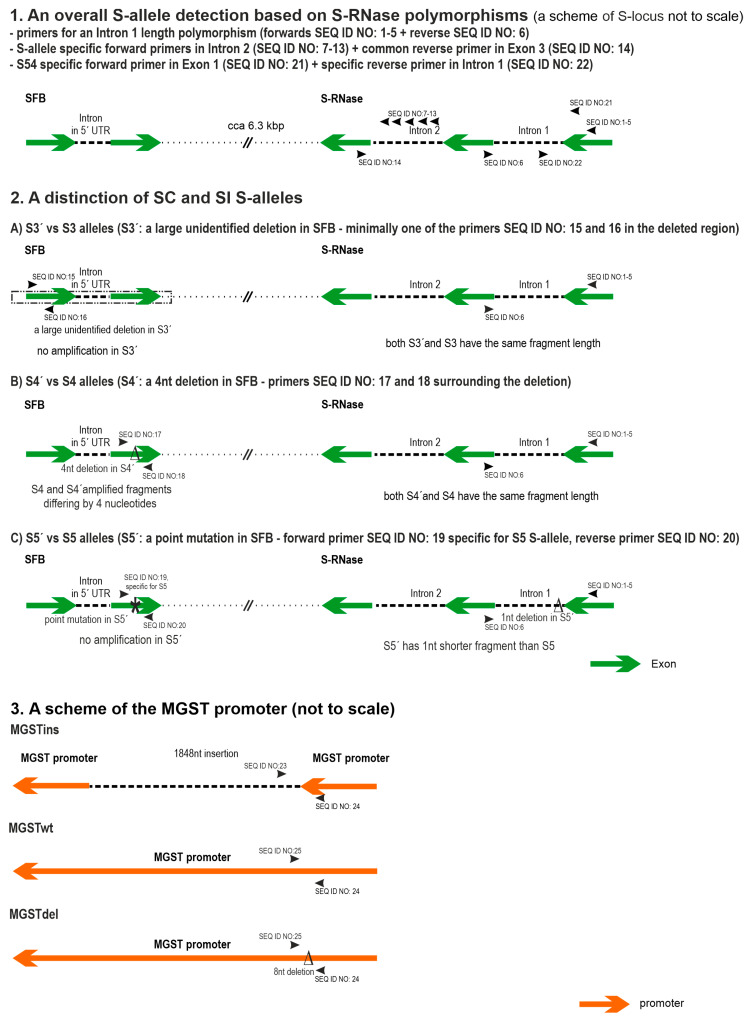
The design of the assay for self-(in)compatibility determination in sweet cherries. The figure is divided into three parts showing: (**1**) overall S-allele detection, (**2**) discrimination of SC and SI S-alleles, and (**3**) MGST allele determination. Drawings based on the ‘Satonishiki’ cultivar reference sequence, BioProject No. PRJDB4877.

**Figure 2 ijms-24-06931-f002:**
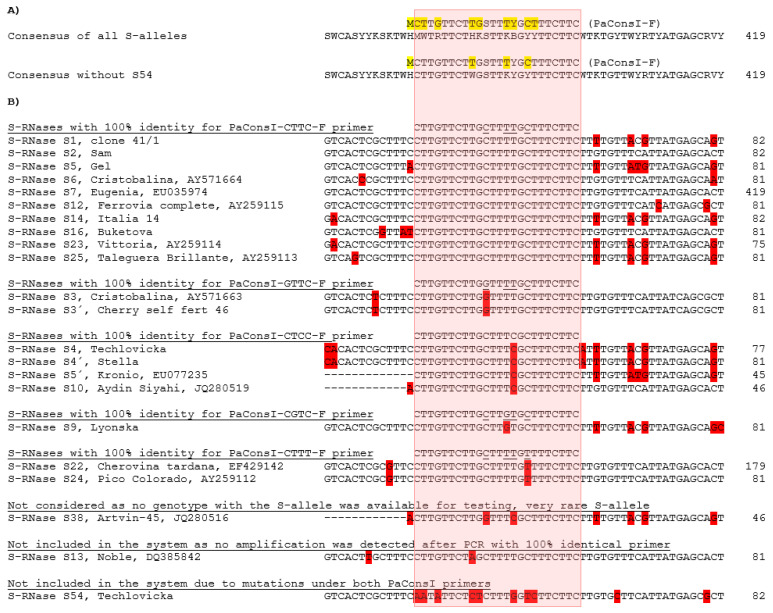
PaConsI-F-class primer binding site sequence analysis in detected S-alleles. The region of the primer is in the pink frame. Where used, the GenBank number is shown next to the genotype name; sequences without a reference were obtained in this study. (**A**) Upper—a consensus sequence of all S-RNases with known sequences, lower—a consensus sequence of all S-RNases with known sequences without the newly identified S54 allele (excluded due to the many mismatches present). The PaConsI-F primer sequence [19] is shown above the consensus. Nucleotides highlighted by yellow in the PaConsI-F primer are mutated in some S-alleles and were not compensated for by a previous degeneration of the primer. (**B**) S-alleles recognized by individual PaConsI-xxxx-F primers. Nucleotides highlighted in red indicate differences compared to the predominant nucleotide.

**Figure 3 ijms-24-06931-f003:**
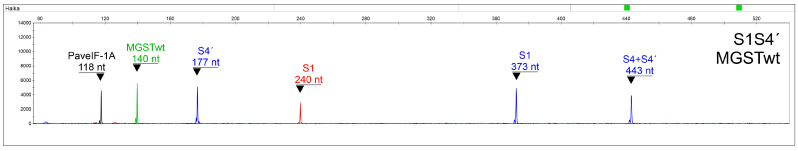
A representative output of fragment analysis in the ‘Halka’ cultivar (S1S4′; MGSTwt).

**Figure 4 ijms-24-06931-f004:**
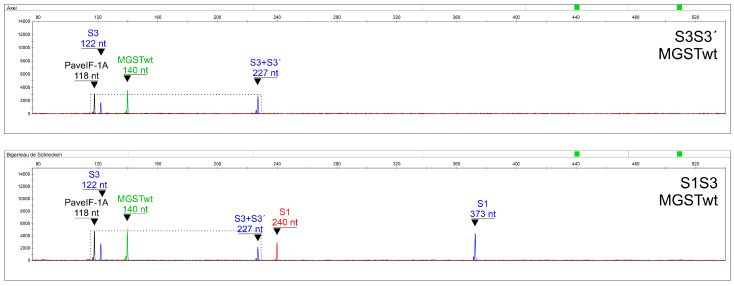
A representative output of fragment analysis for S3S3′ and S1S3 as a prototype of S3Sx genotypes.

**Figure 5 ijms-24-06931-f005:**
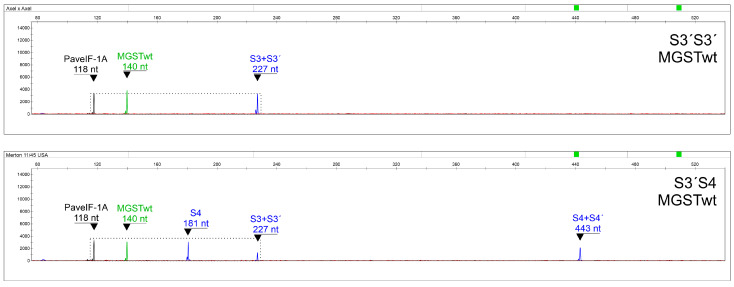
A representative output of fragment analysis for S3′S3′ and S3′S4 as a prototype of S3′Sx genotypes.

**Figure 6 ijms-24-06931-f006:**
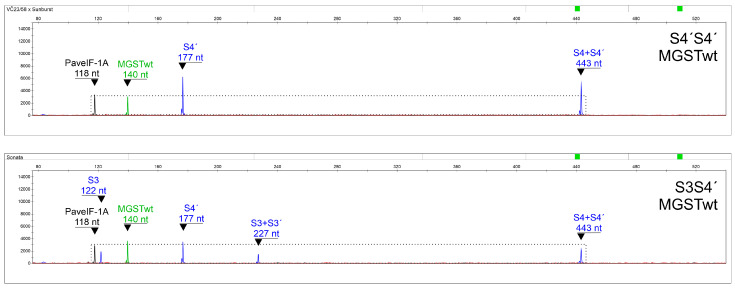
A representative output of fragment analysis for S4′S4′ and S3S4′ as a prototype of S4′Sy genotypes.

**Figure 7 ijms-24-06931-f007:**
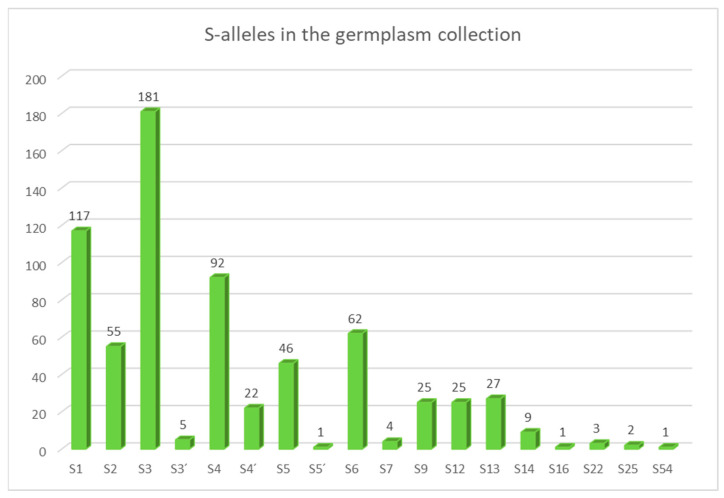
Frequency of individual S-alleles found in the germplasm collection.

**Table 1 ijms-24-06931-t001:** Overview of primers used in the study. MGSTins denotes the allele with an insertion in the MGST promotor described by Ono et al. [15]. MGSTwt stands for the wild-type sequence, and MGSTdel for the 8 nt deletion identified in this study (see Appendix A). Rows with an empty reference cell refer to this study.

SEQ ID NO	Primer Name	Sequence (5′->3′)	Fluorescent Label	Final Concentration (µM)	Alleles Recognized with 100% Homology	Reference
1	PaConsI-CTTC-F	CTTGTTCTTGCTTTTGCTTTCTTC		0.19	S1, S2, S5, S6, S7, S12, S14, S16, S23, S25 S-RNase	
2	PaConsI-GTTC-F	CTTGTTCTTGGTTTTGCTTTCTTC		0.33	S3, S3′ S-RNase	
3	PaConsI-CTCC-F	CTTGTTCTTGCTTTCGCTTTCTTC		0.18	S4, S4′, S5′, S10 S-RNase	
4	PaConsI-CTTT-F	CTTGTTCTTGCTTTTGTTTTCTTC		0.5	S22, S24 S-RNase	
5	PaConsI-CGTC-F	CTTGTTCTTGCTTGTGCTTTCTTC		0.22	S9 S-RNase	
6	PaConsI-R2	GCCATTGTTGCACAAATTGA	FAM	0.21	all S-RNases with a known sequence	[20]
7	S-RNase-S1-In2-F	TGGTCTCCCTAACATGACCC		0.175	S1 S-RNase	
8	S-RNase-S2-In2-F	TGAACGAAATCTCAACTCATAAATC		0.43	S2 S-RNase	
9	S-RNase-S6+S24-In2-F	TCATTTTGTTTTCCACCTACCC		0.18	S6 and S24 S-RNase	
10	S-RNase-S7-In2-F	TCTGTCTGGTTGTTTTGCTGG		0.17	S7 S-RNase	
11	S-RNase-S9+S22-In2-F	TCTAATAATGGATCTGCTCATCTAATT		0.7	S9 and S22 S-RNase	
12	S-RNase-S12-In2-F	GCTAACCCTTACATTTTGACCC		0.25	S12 S-RNase	
13	S-RNase-S13-In2-F	ATATGTCTGTCTATCTATCTGTTTTCTCA		0.4	S13 S-RNase	
14	S-RNases-Ex3-R	GTATCATTGCCACYTTCCACG	PET	0.24	most of S-RNases	
15	PaSFB3-F	CCACAATTTGAACGTCAGAAC		0.28	S3 SFB	[12]
16	PaSFB3-short-R	TCTGTGTTTTCTAAAGGATGGC	FAM	0.28	S3 SFB	
17	PaSFB4+4′-F	TCTAGCTTTTATTCTTGCGAGG	FAM	0.155	S4 and S4′SFB	
18	PaSFB4+4′-R	GATCTCCTATGCCCCTAGAGAA		0.155	S4 and S4′SFB	
19	PaSFB-S5-F	GCTTGGACAAAATTGACTTGTG		0.2	S5 SFB	
20	PaSFB-S5+S5′-R	GATCACAATCACCCAAAGGAGG	FAM	0.2	S5 and S5′SFB	
21	S-RNase-S54-F	CTCTCTTTGGTCTTCTTCTTGTGC	PET	0.17	S54	
22	S-RNase-S54-R	GCTTGCTGATTGTAAATAAACTGC		0.17	S54	
23	MGST-TE-in-F	ATAAATGGGTCAGTGGTGGG		0.105	MGSTins	
24	MGST-TE-out-F	AAAGCCTTCAAGTGGGAAAG		0.105	MGSTwt, MGSTdel	
25	MGST-TE-out-R	TTGCTTACAGGTCATTACTTACACG	VIC	0.105	MGSTwt, MGSTdel, MGSTins	[15]
26	PaveIF-1A-F	GCCCAAGTGCTTCGTATGCT	NED	0.05	only 1 allele of PaveIF-1A observed	[31]
27	PaveIF-1A-R	ATCACCGGCTGCAATCCA		0.05	only 1 allele of PaveIF-1A observed	[31]

**Table 2 ijms-24-06931-t002:** MGST- and S-allele typical fragment combinations. S-alleles with the known sequence in the region amplified by PaConsI-F and PaConsI-R are shown only. PaveIF-1A fragment size is also shown. The number of genotypes containing a respective S-allele is based on a curated list of cultivars (n = 1483; [29]) and is presented to show the coverage of the assay. B: blue detection channel (6-FAM dye); G: green detection channel (VIC dye); Y: yellow detection channel (NED dye); R: red detection channel (PET dye); N.D.: not done. The expected length is based on an existing sequence; observed length refers to a relative fragment size in fragment analysis.

S-allele/MGST Promoter	Length of Fragment #1 (Observed, *Expected*) (nt)	Length of Fragment #2 (Observed, *Expected*) (nt)	Number of Genotypes Described in a Reference List [29]
S1	B: 373 *(376/379)*	R: 240 *(240)*	410
S2	B: 337 *(343)*	R: 302 *(303)*	188
S3	B: 122 *(125)*	B: 227 *(233)*	747
S3′	B: 227 *(233)*		7
S4	B: 181 *(184)*	B: 443 *(450)*	345
S4′	B: 177 *(181)*	B: 443 *(450)*	76
S5	B: 83 *(90)*	B: 385 *(392)*	149
S5′	B: 384 *(391)*		3
S6	B: 435 *(442)*	R: 223 *(225)*	397
S7	B: 339 *(340)*	R: 249 *(250)*	45
S9	B: 350 *(355)*	R: 266 *(266)*	221
S10	N.D. *(363)*		22
S12	B: 338 *(344)*	R: 147 *(148/149)*	73
S13	R: 119 *(121)*		114
S14	B: 323 *(330)*		39
S16	B: 406 *(412)*		47
S21/S25	B: 367 *(374)*		5
S22	B: 415 *(421)*	R: 258 *(253/260)*	39
S23	N.D. *(330)*		0
S24	N.D. *(421)*	R: *(225)*	4
S28	N.D. *(367)*		0
S29	N.D. *(338)*		0
S30	N.D. *(384)*		1
S31	N.D. *(208)*		0
S34	N.D. *(377)*		0
S38	N.D. *(309)*		0
S54	R: 172 *(171)*		0
MGSTwt	G: 140 *(145)*		N.D.
MGSTins	G: 192 *(197)*		N.D.
MGSTdel	G: 132 *(137)*		N.D.
PaveIF-1A	Y: 118 *(120)*		N.D.

**Table 3 ijms-24-06931-t003:** A fragment association with a particular MGST-, S-allele, and the PaveIF-1A control. * Confirmation of the particular S-allele by another fragment. ** According to the alignment of GenBank sequences DQ266445 for S14 and AY259114 for S23, there are only three nucleotides different, and all of them lie in introns. As the S-RNase protein will be the same for both S-alleles, distinguishing these alleles was not a priority of this study.

6-FAM (Blue)		PET (Red)		VIC (Green)		NED (Yellow)	
Observed Length (nt)	S- Allele	Observed Length (nt)	S- Allele	Observed Length (nt)	Allele	Observed Length (nt)	Allele
** 83 **	S5	** 119 **	S13	** 132 **	MGSTdel	118	PaveIF-1A
** 122 **	S3	** 147 **	S12	** 140 **	MGSTwt		
** 177 **	S4′	** 172 **	S54	** 192 **	MGSTins		
** 181 **	S4′	** 223 **	S6				
** 227 **	S3; S3′ *	** 240 **	S1				
** 323 **	S14/S23 **	** 249 **	S7				
** 337 **	S2 *	** 258 **	S22				
** 338 **	S12 *	** 265 **	S9				
** 339 **	S7 *	** 302 **	S2				
** 350 **	S9 *						
** 367 **	S21/S25						
** 373 **	S1 *						
** 384 **	S5′ *						
** 385 **	S5 *						
** 406 **	S16						
** 415 **	S22 *						
** 435 **	S6 *						
** 443 **	S4; S4′ *						

**Table 4 ijms-24-06931-t004:** A comparison of S3S3′ and S3Sx genotypes. Ratios of respective peak heights were calculated and compared. The letter in parentheses indicates the channel a specific peak is detected in Y yellow (represented here by black for contrast); B blue. SD: standard deviation.

	PaveIF-1A (Y): 118 nt/ S3 SFB (B): 122 nt	PaveIF-1A (Y): 118 nt/ S3+S3′ S-RNase (B): 227 nt	S3+S3′ S-RNase (B): 227 nt/ S3 SFB (B): 122 nt
	Average ± SD	Average ± SD	Average ± SD
**S3S3′**	1.75 ± 0.33	1.17 ± 0.21	1.50 ± 0.22
**S3Sx**	1.65 ± 0.26	2.12 ± 0.25	0.78 ± 0.15
** *t* ** **-test (*p*-value)**	0.20	** 2.08 × 10^−22^ **	** 4.77 × 10^−19^ **

**Table 5 ijms-24-06931-t005:** A comparison of S3′S3′ and S3′Sx genotypes. Ratios of respective peak heights were calculated and compared. MGST with only one allele present was used in parallel to show no significant difference between the amplification of the two control genes in both groups. The letter in parentheses indicates the channel a specific peak is detected in Y yellow (represented here by black for contrast); B blue; G green. SD: standard deviation.

	PaveIF-1A (Y): 118 nt/S3+S3′ S-RNase (B): 227 nt	MGST (G): 140 nt/PaveIF-1A (Y): 118 nt	MGST (G): 140 nt/S3+S3′ S-RNase (B): 227 nt
	Average ± SD	Average ± SD	Average ± SD
**S3′S3′**	1.30 ± 0.16	1.14 ± 0.14	1.47 ± 0.16
**S3′Sx**	2.25 ± 0.22	1.10 ± 0.13	2.47 ± 0.27
** *t* ** **-test (*p*-value)**	** 3.48 × 10^−26^ **	0.28	** 6.19 × 10^−22^ **

**Table 6 ijms-24-06931-t006:** A comparison of S4′S4′ and S4′Sy genotypes. Ratios of respective peak heights were calculated and compared. The letter in parentheses indicates the channel a specific peak is detected in Y yellow (represented by a black color for contrast); B blue. SD: standard deviation.

	PaveIF-1A (Y): 118 nt/ S4′(B) 177 nt	PaveIF-1A (Y): 118 nt/S4+S4′ (B): 443 nt	S4+S4′ (B): 443 nt/S4′(B) 177 nt
	Average ± SD	Average ± SD	Average ± SD
**S4′S4′**	0.53 ± 0.06	0.66 ± 0.08	0.81 ± 0.05
**S4′Sy**	1.04 ± 0.13	1.34 ± 0.13	0.78 ± 0.05
** *t* ** **-test (*p*-value)**	**7.44** ** × 10 ** ** ^−22^ **	**2.76** ** × 10 ** ** ^−28^ **	0.052

## Data Availability

Not applicable.

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
