# Peer review of "A New One-Tube Reaction Assay for the Universal Determination of Sweet Cherry (Prunus avium L.) Self-(In)Compatible MGST- and S-Alleles Using Capillary Fragment Analysis"

_ijms, 2023, doi:10.3390/ijms24086931_

Round 1
Reviewer 1 Report
Dear Editor
International Journal of Molecular Sciences
I am composing my feedback on a manuscript that has been presented to the International Journal of Molecular Sciences under the identification number ijms-2297541. The article outlines a novel method for identifying self-(in)compatible MGST- and S-alleles in sweet cherry (Prunus avium L.) using capillary fragment analysis in a single-tube reaction assay.
The authors have introduced a straightforward method for detecting multiple S-alleles and MGST promoter variants using a single-tube PCR and fragment analysis on a capillary genetic analyzer. They have successfully identified three MGST alleles, 14 self-incompatible S-alleles, and all three known self-compatible S-alleles (S3', 19 S4', S5') in 55 combinations tested. The authors suggest that this method is ideal for routine S-allele diagnostics and molecular marker-assisted breeding for self-compatible sweet cherries. Although there are some minor concerns that need clarification, we believe that the data presented in this study are valuable and informative, making it a potential candidate for publication.
1. Throughout the text, please replace the term 'variety' with 'cultivar'.
2. Figure 1 accurately uses the term 'polymorphism'.
3. The introduction is too lengthy and should be condensed.
4. Identifying S-alleles is a complex process, and using different DNA extraction methods and PCR reaction mixtures may not yield consistent results. The authors used one method for both DNA extraction and PCR reactions, but it is unclear if this system will work with other methods or conditions from different companies. Therefore, I suggest that the results be verified using alternative DNA extraction methods and PCR reaction mixtures from other sources.
5. The writers claim that their approach reduces the cost of analysis, but this assertion may not be accurate as the primers require labeling and a genetic analyzer, which may not be accessible in all locations and necessitate additional expenses for external services. Therefore, it is advisable to reassess the cost-effectiveness of their method.
6. Could you make the conclusion section more inclusive and thorough?
Ultimately, I believe that the current draft of this manuscript meets the expected level of quality for publication in the International Journal of Molecular Sciences. However, significant revisions are necessary before it can be deemed acceptable for publication.
Reviewer 2 Report
I think the rationale for this study is very good-they want to have a single reaction assay that recognizes the alleles of the MGST and the S-alleles in cherry. I found this very difficult to figure out how the average cherry breeder would set these reactions up (methods are unclear about which primers to select) and some of the figures were unclear and had spelling errors.
Keywords: self-incompatibility is listed twice
Introduction:
Line 34-define 'mutually pollinated'
Line 34-35: rewrite sentence for clarity
Line 46: delete 'in essence'
Line 151-replace 'top' with 'improved'. What traits are they improving?
Figure 1-was difficult to follow. The word 'polymorphisms' and 'length' are misspelled throughout. For the distinction of S3' and S5' how can you tell if it has this allele or it's just PCR failure? In C) no reverse primer listed in title. For section 3 of Figure 1: 'not to scale'
Line 190-191: signal height seems a bit dangerous to decide which alleles you have as it can be impacted by PCR efficiency and how much template you add.
Table 1-Are all these primers being used in one reaction?? I would suggest breaking the paper into 2 sections- one where you worked on developing the methods and the second section would be using the methods. At this time it's hard for me to figure out how you are setting up your reactions
Figure 2-Did you submit the sequences you genrated to GenBank?
Line 546: replace 'huge' with 'large'
Line 546-549: Does Washington State use this money to select for S-alleles or for other traits as well?
Line 579-Did you really extract DNA from the phloem?
Line 618-How many primers are you using? At what volume and concentration?
Round 2
Reviewer 2 Report
The authors addressed most of my comments and clarified that indeed they are using 27 primers for each reaction. 27 primers is a tremendous number of primers so I suggest removing the word "simple" to describe this new method. Also I'm still unclear of how they setup the reactions/thermocycler conditions when they used Phusion polymerase for all 27 primers. I see a description of using Phusion but it appears to be when they were working out some PCR conditions for certain primers. Please clarify.
The discussion where they talk about Washington University: please revise the sentence where you use "provided resources surplus"? This is an odd fragment and I'm not sure what this means.
